# Exploring the Role of microRNAs in Glioma Progression, Prognosis, and Therapeutic Strategies

**DOI:** 10.3390/cancers15174213

**Published:** 2023-08-22

**Authors:** Omar Tluli, Mazyona Al-Maadhadi, Aisha Abdulla Al-Khulaifi, Aishat F. Akomolafe, Shaikha Y. Al-Kuwari, Roudha Al-Khayarin, Cristina Maccalli, Shona Pedersen

**Affiliations:** 1College of Medicine, Qatar University, Doha P.O. Box 2713, Qatar; ot2004691@qu.edu.qa (O.T.); ma1806279@qu.edu.qa (M.A.-M.); aa1901291@qu.edu.qa (A.A.A.-K.); aa1905814@qu.edu.qa (A.F.A.); ra1702398@qu.edu.qa (R.A.-K.); 2Sidra Medical and Research Center, Ar-Rayyan P.O. Box 26999, Qatar; cmaccalli@sidra.org

**Keywords:** microRNA, glioma, cancer stem cells, prognosis, targeted therapy

## Abstract

**Simple Summary:**

Despite advancements in healthcare and research, the occurrence of gliomas, a type of brain tumor, continues to rise. Emerging evidence has proven that dysregulated microRNAs play a significant role in the initiation, progression, prognosis, and recurrence of gliomas. Not only can these micro-RNAs serve as diagnostic tools, but they also hold promise for the development of targeted therapeutic treatments. It is therefore of great importance to have a comprehensive understanding of the specific microRNAs involved, the different pathways they are involved in, and the potential outcomes of mutations. This is ultimately the focus of this review, which establishes a solid foundation for the development of targeted therapeutic agents, also highlighting the possible challenges that may be encountered.

**Abstract:**

Gliomas, which arise from glial cells in the brain, remain a significant challenge due to their location and resistance to traditional treatments. Despite research efforts and advancements in healthcare, the incidence of gliomas has risen dramatically over the past two decades. The dysregulation of microRNAs (miRNAs) has prompted the creation of therapeutic agents that specially target them. However, it has been reported that they are involved in complex signaling pathways that contribute to the loss of expression of tumor suppressor genes and the upregulation of the expression of oncogenes. In addition, numerous miRNAs promote the development, progression, and recurrence of gliomas by targeting crucial proteins and enzymes involved in metabolic pathways such as glycolysis and oxidative phosphorylation. However, the complex interplay among these pathways along with other obstacles hinders the ability to apply miRNA targeting in clinical practice. This highlights the importance of identifying specific miRNAs to be targeted for therapy and having a complete understanding of the diverse pathways they are involved in. Therefore, the aim of this review is to provide an overview of the role of miRNAs in the progression and prognosis of gliomas, emphasizing the different pathways involved and identifying potential therapeutic targets.

## 1. Introduction

Brain malignancies are among the most dreaded forms of cancer because of their immediate impact on cognitive function and well-being and unfavorable prognosis. These tumors exhibit irregular growth patterns and invade surrounding healthy brain tissue. The location and size of the tumor can lead to a variety of symptoms, including headaches, seizures, numbness, and difficulties with speech and vision. Glial cells, which provide support and protection to neurons, can transform into gliomas, the most prevalent type of primary malignant brain tumor [1].

Gliomas comprise various subtypes, including oligodendrogliomas, astrocytomas, and ependymomas, representing 24% of all primary brain and CNS malignancies worldwide. The incidence of gliomas differs depending on histology, pilocytic astrocytomas being more prevalent in children and adolescents, low-grade oligodendrogliomas peaking between the ages of 30 and 40, and glioblastoma (GBM) rising in people in their 60s and 70s [2]. A study on the prevalence and patterns of tumors in the brain and central nervous system revealed a considerable rise of 94.35% in cases between 1990 and 2019 [3]. A similar study found a 152.5% increase in cases in North African and Middle Eastern countries over the same period [4]. 

According to the World Health Organization (WHO) classification system, gliomas are classified into four types based on their histopathological characteristics, reflecting the tumor’s aggressiveness and propensity for rapid growth and dissemination. Low-grade gliomas, which grow slowly, often have a better prognosis than high-grade gliomas. Such gliomas can originate from any glial cell and are categorized as either grade I or II, with examples including pilocytic astrocytoma and oligodendroglioma. Meanwhile, high-grade gliomas, often referred to as malignant gliomas, are a group of aggressive, rapidly proliferating tumors that fall into two classifications: grade III anaplastic gliomas and grade IV glioblastomas. Although the precise etiology of gliomas is unknown, numerous genetic abnormalities and mutations have been shown to contribute to the growth and development of these tumors. For example, mutations in the isocitrate dehydrogenase 1 and 2 (IDH1, IDH2) genes, which typically play essential roles in cellular metabolism, particularly in the citric acid cycle, are frequently observed in low-grade gliomas. In contrast, tumor protein 53 (TP53), phosphatase and tensin homolog (PTEN), and epidermal growth factor receptor (EGFR) gene mutations are frequently found in high-grade gliomas [5]. Consequently, gliomas are challenging to diagnose and treat, and current treatments such as surgery, radiotherapy, and chemotherapy are frequently ineffective. 

Recent studies suggest that cancer stem cells (CSCs) play a critical role in the development, progression, and recurrence of gliomas. CSCs are a subpopulation of tumor cells with self-renewal ability that can differentiate into various cell types and are found to be resistant to conventional therapies, including chemotherapy, immunotherapy, and radiotherapy [6]. A major contributor to the characteristic properties of CSCs, also seen in gliomas, is the dysregulation of miRNAs. By modulating the expression of key genes and signaling pathways involved in the maintenance and differentiation of CSCs, abnormal miRNA can lead to the accumulation and expansion of CSCs, thereby promoting tumor growth and resistance to therapy [7]. This provides hope for developing innovative cancer treatments based on an understanding of the molecular mechanisms of cancer. 

MiRNAs are small-in-length (from 21 to 25 nucleotides), non-coding RNA molecules crucial in regulating gene expression. They are found in many organisms, including plants, animals, and viruses. They, through binding to regions of messenger RNA (mRNA) molecules, exert post-translational regulatory effects on genes. MiRNAs can regulate gene expression by binding to the specific areas of mRNAs, causing them to either degrade or experience inhibited translation, ultimately preventing protein synthesis. Likewise, long non-coding RNAs (lncRNAs) share similar characteristics, distinguished by their extended nucleotide sequences, where they also actively participate in the intricate regulation of gene expression across multiple levels. In fact, they are found to interact with miRNAs, an interaction that is highly consequential, giving rise to complex regulatory networks that play pivotal roles in gene expression and various cellular processes. Accordingly, miRNAs are associated with many biological processes, including development, differentiation, apoptosis, and immune responses [8]. Their dysregulated expression has been implicated in numerous diseases, including cancer, cardiovascular disease, and neurological disorders. They can be used as diagnostic and prognostic biomarkers for various diseases and as potential targets for therapeutic intervention.

MiRNAs are incapable of being translated into proteins; instead, they bind to the 3’ untranslated region (UTR) of the target messenger RNA (mRNA), influencing its stability and/or translation. Approximately 33 percent of the human-expressed mRNAs involved in cell development, differentiation, and death are regulated by miRNAs. 

Multiple miRNAs, including miR-21, miR-10b, miR-128, miR-34a, and others, are dysregulated in glioblastoma [9]. These miRNAs regulate various pathways that contribute to the development and progression of glioblastomas, such as cell proliferation, invasion, and apoptosis. MiR-21, for example, has been shown to promote tumor growth and invasion by targeting various tumor suppressor genes when overexpressed in glioblastoma. By targeting the HOXD10 gene, miR-10b has also been shown to promote glioblastoma cell migration and invasion. The dysregulation of miRNAs in glioblastoma suggests that they could be potential targets for therapeutic targets. Strategies targeting specific miRNAs have been developed, such as miRNA replacement therapy and miRNA inhibition therapy, and are being investigated as potential treatments for glioblastoma.

Overall, the involvement of miRNAs in the advancement and growth of glioblastoma is significant, providing opportunities for novel therapeutic approaches. This review aims at providing an overview of the potential application of miRNAs in determining the development and prognosis of GBM.

## 2. Method for Article Search and Selection

The aim of our research was to explore the role of microRNAs in gliomas, focusing on their contribution to glioma progression, prognosis, and treatment. To achieve this, we conducted a thorough search using reputable academic databases, including PubMed, Web of Science, and Scopus. Our search strategy involved combining specific keywords to narrow down relevant articles. The search terms included “microRNA” or “miRNA”, “glioma”, “glioblastoma”, “progression”, “prognosis”, “treatment”, “therapeutic strategies”, and “cancer stem cells”. We included (1) articles published in peer-reviewed journals to ensure reliability and scientific validity, (2) studies that presented scientific evidence, including pre-clinical or clinical research on microRNAs, (3) articles focusing on specific microRNAs and their normal pathways, (4) papers discussing the involvement of microRNAs in glial brain tumors, and (5) studies providing recent insights into microRNA-dependent modulation of gliomas, encompassing aspects of pathogenesis, diagnosis, and treatment. We excluded (1) studies not published in peer-reviewed journals to maintain high research quality, (2) articles not directly related to gliomas or microRNAs, which might not align with our research focus, (3) papers not concentrating on microRNA regulation or cell cycle/apoptosis mechanisms in glioma, (4) articles published before 2009 to focus on recent developments in the field, and (5) articles not written in English to ensure accessibility and understanding. By applying these criteria, we aimed to select articles that comprehensively address the role of microRNAs in gliomas, providing valuable insights into their potential as therapeutic targets.

## 3. The Role of miRNAs in Gliomas

Numerous miRNAs participate in cancer development through cell cycle regulation, apoptosis, and DNA damage response pathways. As mentioned earlier, there is mounting evidence that various miRNA mutations may play a role in the advancement and growth of glioblastomas [10]. Depending on whether they are under-expressed or over-expressed, microRNAs can have a dual effect on the cell cycle by both positively regulating survival pathways and disrupting them (Table 1). 

### 3.1. MiRNA-21,221 and 222

One of the principal mechanisms of cancer cell survival is the avoidance of programmed cell death. This can be accomplished by dysregulating various miRNAs, including miRNA-21, miR-221, and miR-222 [10]. MiRNAs can exhibit either pro-apoptotic or anti-apoptotic characteristics. Those that target genes that promote apoptosis are frequently observed in glial tumors, and it is believed that miR-21, due to its anti-apoptotic nature, may have micro-oncogenic properties [23]. In GBM cells and malignant brain tissues, miR-21 expression is significantly elevated. It modulates P53 and TGF-beta activity, consequently decreasing apoptosis [11]. Furthermore, miR-21 can specifically suppress Fas ligand in tumor stem cells, inhibiting apoptosis [12]. Predictably, gliomas demonstrated increased sensitivity to chemotherapy upon the inhibition of miR-21 [12]. Similarly, both miR-221 and miR-222 exhibit anti-apoptotic properties through their interaction with PUMA (p53 upregulated modulator of apoptosis), which typically controls apoptosis. In gliomas, an increased miR-221/222 level has been observed to decrease PUMA expression. This promotes the survival and growth of tumor cells [13].

### 3.2. MiR-296 and 93

Angiogenesis, a critical process for tumor survival, is affected by the abnormal expression of these miRNAs. In vitro studies have shown that excess VEGF, a proangiogenic factor, can enhance the endogenous production of miR-296 in human glial cells [24]. In a feedback loop, miR-296 promotes the growth of new blood vessels by upregulating the expression of the vascular endothelial growth factor receptor 2 (VEGFR2) [14]. The upregulation of miR-93 can suppress integrin-β8, a protein that participates in cell and cell–matrix interactions, consequently promoting angiogenesis. Studies have demonstrated that when human U87 glioblastoma cells are co-cultured with endothelial cells and miR-93 is overexpressed, there is a notable increase in both the proliferation and formation of tube-like structures by endothelial cells [15].

### 3.3. MiRNA-138 and MiRNA-490

These miRNAs have tumor-suppressive properties and inhibit glioblastoma tumorigenicity [16]. It was noted that there is an inverse relationship between the genetic expression of MiRNA-138 and the levels of CDK6, representing a crucial regulator of the G1 to S transition phase of the cell cycle that is usually increased in GBM [17]. MiRNA-490 has been identified to function as a tumor suppressor in GBM, and glioma cell lines have significantly lower levels of this miRNA. It has been shown to inhibit HMGA2, an oncogenic protein, and TERF2, a protein responsible for telomere maintenance. MiRNA-138 and MiRNA-490 are downregulated in GBM, promoting tumorigenesis [18]. 

### 3.4. MiRNA-128

MiRNA-128 is one of the most frequently detected miRNAs in the brain. It is crucial for CNS cell development and maturation and neuronal maturation [25]. It also regulates the migration of neurons in the cerebral cortex [26]. In addition, it suppresses tumors by inhibiting the oncogenes PRC and BMI1, resulting in the reduced proliferation of glioblastoma stem cells [19]. Loss of MiRNA-128 occurs early in the formation of gliomas, leading to reduced differentiation and increased stemness, giving rise to the growth of more aggressive tumors [20].

### 3.5. MiRNA-124 and MiRNA-137

MiRNAs 124 and 137 are involved in determining the fate and differentiation of neurons. The growth factors EGF and PDGF have been shown to reduce the expression of these miRNAs [20]. The upregulation of miRNA-124 promotes the neuronal differentiation of progenitor cells, encourages the generation of neuroblasts and mature neurons, and decreases the invasiveness of GBM [20,21]. It also downregulates Sox9, which usually promotes the proliferation of precursor cells [27]. MiRNA-124 is typically absent or underexpressed in GBM, resulting in tumor expansion and the division of glioblastoma stem cells [28]. Similarly, miRNA-137 plays a role in the maturation of immature neurons [22]. It also inhibits the ability of glioblastoma stem cells to self-renew by suppressing LSD1 and TLX, which generally keep stem cells undifferentiated and capable of self-renewal [29].

## 4. miRNAs and Cancer-Associated Pathways in Gliomas

In GBM, miRNAs act on pre-altered genetic pathways, such as the wingless/integrated (WNT) pathway, P53 pathway, and retinoblastoma (RB) pathway, leading to increased proliferation rate, decreased apoptosis, angiogenesis, de-differentiation, and metastasis [23].

### 4.1. WNT Pathway

The WNT signaling pathway is vital in determining cell fate and regulating cell migration, cell polarity, and neural patterning. In addition to its normal function, the WNT pathway can also be activated abnormally in a variety of cancers, including melanoma, ovarian cancer, colorectal cancer, breast cancer, and prostate cancer. For example, the WNT pathway and the transcription factors associated with it have a significant impact on glioma growth and progression [30]. Furthermore, activation of this pathway promotes the maintenance of GMB stem cells, and the resulting increase in self-renewal capacity is initiated by WNT signaling regulators such as PLAGL2, forkhead box protein M1 (FoxM1), and achaete-scute homolog (ASCL1) [31]. Multiple WNT pathways have been identified, including a canonical WNT pathway and several non-canonical WNT pathways. The canonical WNT pathway is also known as the β-catenin-dependent WNT pathway because it involves the interaction of WNT proteins and β-catenin. However, non-canonical WNT pathways do not depend on β-catenin for their signaling; these pathways can be further classified into the Planar Cell Polarity (PCP) and WNT-Ca^2+^ pathways. The two pathways mentioned above are known to be linked with and involved in different cellular processes, but both, when abnormally induced, have been related to cancer development [32].

Furthermore, the WNT pathway and miRNAs have a symbiotic relationship in cancer. The hyperactivation of the WNT pathway can be caused by miRNA dysregulation. MiRNA expression is monitored and controlled by the WNT signaling pathway [33].

The binding of WNT ligands to various cell receptors, including LRP 5/6, frizzled receptor (FZD), protein tyrosine kinase 7 (PTK7), RAR-related orphan receptor (ROR), or receptor-like tyrosine kinase (RYK) activate both canonical and non-canonical WNT signaling pathways. To date, 19 WNT ligands have been identified, with specific ligands (WNT1, 2, 2B, 3, 3A, 7A, 7B, 9A, 9B, 10A, and 10B) primarily associated with the canonical pathway [33]. On the other hand, WNT7A and WNT7B are mainly linked to the WNT/PCP pathway, while WNT5A and 5B are connected to the WNT/Ca2+ pathway, but they also play a role in the canonical pathway [34]. In addition, research has indicated that ligands such as WNT1, 2, 2B, 3A, 5A, 6, 7A, and 7B are implicated in the progression of gliomas [35].

MiR-122 targets WNT1 directly and inhibits the canonical WNT pathway and an increase in the levels of phosphorylated β-catenin, whereas other miRNAs, such as miR-133b, miR-362-3p, and miR-139-5p, target WNT1 indirectly and suppress the WNT/β-catenin pathway [36]. In contrast, TRIM24 activates the WNT pathway and other signaling pathways in gliomas, such as the EGFR and STAT3 pathways. Furthermore, TRIM24 expression is negatively regulated by miR-138-2-3p, suppressing the protein levels of WNT1 and WNT3a. Moreover, lncRNA NCK1-AS1 acts as a “sponge” for miR-138-2-3p, activating the WNT pathway. This suggests that NCK1-AS1 binds to miR-138-2-3p, deterring it from interacting with its target genes and increasing WNT pathway activity [33]. 

### 4.2. P53 Pathway

It was revealed that P53 is both a tumor suppressor and transcription factor; it is involved in many biological functions, including cell cycle arrest, genome stability, programmed cell death, regulation of cellular metabolism, and inhibition of the growth of new blood vessels [34]. The P53 pathway comprises MDM2, ARF, CDKN2A, and P53, among which P53 is a critical regulator. This pathway is frequently abnormal in more than 50% of GBM patients [37]. Possible P53 pathway changes include the loss of CDKN2A/ARK and the augmentation of MDM2 and MDM4. Furthermore, in gliomas, P53, MDM2, and ARF are frequently mutated, resulting in uncontrolled cell growth and compromised DNA repair. PTEN tightly regulates P53 through a negative feedback loop, denoting that an increase in PTEN expression can lead to a loss of P53 function, even if P53 is not mutated. Mutations in both genes are required for GBM progression and evasion of cell death [11]. Moreover, the overexpression of particular miRNAs, such as miR-21 and miR-26, impacts this pathway and contributes to the development of GBM [38].

### 4.3. Retinoblastoma (RB) Pathway

Retinoblastoma (RB) is a phosphoprotein that suppresses the process of cell division during the G1-S phase, and its activity is controlled by cyclin-dependent kinases (CDKs) with their cyclin associates Cdk4, Cdk6, and Cyclin E-CDK2 [39]. MiRNAs such as miR-124, miR-137, miR-34a, and miR-128 also regulate the RB pathway. MiR-124 and miR-137 downregulation is linked to gliomas, whereas their overexpression can increase cell cycle blockage. Furthermore, by targeting CDKs and transcription factors, these miRNAs can act as tumor suppressors, increasing cell cycle suppression and cell proliferation inhibition [11].

## 5. MiRNAs as Therapeutic Targets for Gliomas

MiRNAs can be expressed differently in normal and malignant tissues, with a possible role in regulating tumor development, invasion, and metastasis. As a result, the aberrant level of expression of miRNAs can worsen the fate of gliomas. In addition, miRNAs are secreted in the cerebrospinal fluid (CSF) and bloodstream. They can freely migrate between normal and tumor cells, making them a potential diagnostic and prognostic biomarker for gliomas [23]. 

One way to categorize miRNAs in cancer is based on their regulator function: oncogenic miRNAs (oncomiRs) (Table 2) and tumor suppressor miRNAs (Table 3). In miRNA-based therapy, either the inhibition or the mimicking of defined miRNAs depending on their expression levels, is utilized [40].

The primary treatment for most central nervous system tumors involves maximally safe surgical resection, which enables accurate histopathological diagnosis, tumor genetic testing, and a reduction in tumor size. This is followed by radiotherapy with concurrent daily temozolomide (TMZ) and an additional six cycles of maintenance of TMZ. The medial overall survival for radiotherapy + TMZ was 14.6 months, whereas in patients who received radiotherapy had a median survival rate of 12.1 months [52]. Unfortunately, when treating relapsed glioblastoma, the available options are not well established, and, despite several ongoing investigations, till this day, none have provided conclusive results to support any interventions that could extend overall survival. Moreover, a considerable proportion of patients may not meet the criteria for second-line therapy. Therefore, some options include further surgical resection, reirradiation, or systemic treatments like lomustine or bevacizumab [53]. Most studies on macrophages and microglia have found that the upregulation of many genes in GAMs, glioma-associated microglia, and macrophages, is inconsistent with support for GBM progression. Molecular targets offer M1 and M2 identifiers, the pro-inflammatory and anti-inflammatory macrophage subtypes, respectively, in a mixed manner. Despite this debate, some typical M2 options are available for therapeutic intervention. However, studies investigating macrophage polarization markers that encourage tumor growth and using macrophage-targeted therapies for GBM treatment have not yet been conducted in zebrafish models [54]. 

Most tumors exhibit dysregulated miRNA expression patterns, and approximately 50% of miRNA-encoded human genes are located in genetic regions associated with cancer. These alterations often lead to a decrease in tumor suppressor genes or an increase in oncogenes, promoting tumor growth [54]. The overexpression of miRNAs such as miR-7, miR-34a, miR-128, miR-124, miR-137, and miR-181 has been observed to impact GBM growth negatively. In contrast, miR-21, the first oncomiR to be studied, plays a crucial role in detrimental processes associated with GBM by targeting the genes mentioned previously and other genes involved in cell proliferation, survival, and drug resistance. MiRNA-21 is one of the many up-regulated miRNAs that have been identified. Likewise, other miRNAs that are also found to be upregulated in GBM and play pivotal roles in glioma genesis are the miR-17-92 cluster, miR-10b, miR-15b, miR-26a, miR-93, miR-148, miR-182, and miR-221/222 [54]. Furthermore, by explicitly targeting vital proteins or enzymes involved in metabolic pathways such as glycolysis, oxidative phosphorylation, glutamine, glucose, and lipid metabolism, various miRNAs can facilitate the metabolic alterations common in GBM [55]. For example, miR-106a and miR-143 target the GLUT-3 transporter and inhibit PKM2 expression, affecting glycolysis in GBM. This same outcome is achieved by targeting HK-2 and let-7a by miR-326, which also inhibits PKM2 expression. MiRNAs that disrupt the function of mitochondria and energetic homeostasis include Lrt-7, miR-16, and miR-23. Other miRNAs in GBM regulate mitochondrial proteins such as ATP5A1 and ATP5B. Given that GBM tumors are highly dependent on aerobic glycolysis, the ability of specific miRNAs to regulate oncogenes and tumor suppressor genes in RTK pathways and their secondary effector pathways is crucial for treating diseases [56]. 

Despite the standard treatment for GBM, which includes surgery, radiotherapy, and TMZ, a subsequent recurrence of GBM is very common, and the average survival time is only 12 to 15 months because of the following problems: -High potential for invasion and proliferation, making it challenging to obliterate all tumors.-High mutational capability, which quickly promotes chemotherapeutic drug resistance, like that to TMZ.

The treatment strategies aimed at targeting and reprogramming tumor-associated macrophages (TAMs) towards M1 anti-tumor macrophages are notably intriguing. These include approaches that alter the tumor microenvironment or inhibit angiogenesis. In experimental murine models, for instance, intertumoral delivery of IL-12 using a modified genetic virus, either alone or in conjunction with immune checkpoint inhibitors (ICIs), demonstrated promising outcomes. Several strategies for genetic, epigenetic, and metabolic remodeling of GBM tumors are being investigated. Radiotherapy, for example, showed anti-tumor efficacy when used alone or combined with histone deacetylase inhibitors and shRNAs against HDAC1 and SIRT1. MiRNA modification is an appealing target because it has the potential to either kill cancer cells or reprogram the TME [54]. 

## 6. Potential Challenges Associated with Targeting miRNAs in Gliomas

While targeting miRNAs appears to be a significant breakthrough in glioma therapy, various challenges hinder its clinical application from becoming a reality in the near future. A notable characteristic of miRNAs is their ability to affect a vast network of proteins by targeting multiple mRNAs, while numerous miRNAs can influence a single mRNA. This can be explained by the degeneracy of miRNA target recognition; the binding site of a miRNA is only partially complementary to the target mRNA, allowing for some mismatch and resulting in low specificity of miRNAs [57]. Simultaneously, competition may emerge between miRNAs and other factors for binding sites on specific mRNAs. As a result, targeting a single miRNA could have unintended consequences to the point where other pathways might counteract the desired therapeutic effect [58].

Furthermore, as a result of alternative splicing, different 3′ untranslated regions (UTRs) with different miRNA binding sites may exist, allowing a single miRNA to target many gene isoforms, resulting in divergent regulation [59]. To demonstrate how the diversity of 3′ UTRs impedes miRNA therapy, it was shown that several genes involved in fundamental physiological functions resist miRNA control due to a lack of miRNA binding sites to their short 3′ UTRs [60]. In addition, miRNA diversifies even more during embryonic development due to the extension of 3′ UTRs via alternative polyadenylation [61]. 

MiRNAs’ ability to target multiple mRNAs can lead to challenges in both delivering miRNA therapies and directing them to specific targets, especially under stressful conditions like those found in gliomas. Moreover, miRNAs may potentially relocate across different intracellular compartments, complicating therapy delivery [62].

Gliomas possess highly heterogeneous microenvironments with varying physiological properties, such as blood flow, throughout the tumor. This can result in inconsistent treatment outcomes due to the altered availability and potency of miRNA therapies [63]. This heterogeneity can ultimately cause gliomas to develop resistance to miRNA-targeting agents, limiting their effectiveness. Furthermore, gliomas’ diverse genomic and phenotypic properties underly the complexity of identifying defined the consistently dysregulated miRNAs across various glioma types.

Crosstalk among signaling pathways in gliomas also complicates the identification of specific miRNAs driving tumor growth. For instance, gliomas’ commonly activated PI3K/AKT and MAPK/ERK pathways promote cell survival, proliferation, and migration. However, their interaction makes it challenging to pinpoint specific miRNAs as therapy targets and contributes to gliomas becoming more resistant [64]. This highlights the need to understand the complex interplay between different pathways in developing effective treatment options.

Lastly, clinical trials for miRNA-based therapeutics have not yet been conducted due to prolonged and challenging regulatory clearances and substantial investment requirements.

## 7. Conclusions

Dysregulated miRNA expression has been connected to tumor progression and unfavorable prognosis in glioma patients. This review highlights the numerous studies pinpointing specific miRNAs associated with glioma progression and prognosis. For instance, miR-21, miR-10b, and miR-221/222 have been identified as upregulated in glioma tissues, correlating with tumor aggressiveness and poor patient survival. Conversely, miR-7, miR-128, and miR-124 have been observed as downregulated in glioma tissues, relating to improved prognosis. Despite this, specific miRNAs can serve as diagnostic markers for glioma, and preclinical studies indicate that miRNA-targeting therapeutic interventions hold promise. Nevertheless, certain miRNAs can potentially act as diagnostic indicators for glioma, and preliminary studies imply that therapeutic strategies targeting miRNAs show potential.

## Figures and Tables

**Table 1 cancers-15-04213-t001:** The role of miRNAs in gliomas.

MiRNA	Function	Expression in Gliomas as Compared to Normal
MiRNA 21	Anti-apoptotic; encourages chemoresistance [11,12].	Increased
MiRNA 221/222	Anti-apoptotic; promotes tumor cell survival and growth [13].	Increased
MiRNA 296	Promotes angiogenesis via interaction with VEGF and VEGFR2 [14].	Increased
MiRNA 93	Angiogenesis: induces the formation of new blood vessels, promotes the increased proliferation of endothelial cells [15].	Increased
MiRNA 138	Tumor suppressor; regulates the cell cycle regulator CDK6 [16,17].	Decreased
MiRNA 490	Tumor suppressor; inhibits oncogenic protein [18].	Decreased
MiRNA 128	Reduces stem cell proliferation and stemness [19,20].	Decreased
MiRNA 124	Neuronal differentiation of progenitor cells; generation of neuroblasts and mature neurons; decreases the invasiveness of GBM [20,21].	Absent or decreased
MiRNA 137	Maturation of immature neurons; reduces the self-renewal capacity of glioblastoma stem cells [22].	Decreased

**Table 2 cancers-15-04213-t002:** Oncogenic miRNAs (oncomiRs) in gliomas.

miRNA	Functions and Regulation	Type of Glioma
miR-21	Anti-apoptotic factor targeting p53 network and TGF- β [41].	Glioblastoma
miR10b	Commonly upregulated in glioblastoma located in HOX cluster. Poor prognosis. Proliferation, migration, and invasion. [41,42]	Glioblastoma
miR-128	Associated with glioma stem cell properties and downregulated. Neuronal differentiation via Bim-1 and EGFR signaling pathways are found to be direct targets. Tumorigenesis regulation. DNA methylation causes upregulation and is a potential target for the upregulation of miR-128-1. [41,43]	Glioblastoma
miR-196a and miR-196b	Highly expressed and shows a significant association with overall survival. Associated with the malignant transformation of gliomas. Poor prognosis by promoting cellular proliferation via cell cycle interference. [41,44].	Glioblastoma

**Table 3 cancers-15-04213-t003:** Tumor suppressor miRNAs in gliomas.

miRNA	Functions and Regulation	Type of Gliomas
miR-15a, miR-16	Downregulation of miR-15a expression: adverse prognosis of human glioma. Upregulation of bcl-2 (anti-apoptotic gene). miR-16- upregulation: mediates glioma growth and invasiveness. [28,45].	Astrocytoma
miR-7	Functions such as survival, proliferation, apoptosis, angiogenesis, and invasion. Targets multiple oncogenes such as PI3K and Raf-1 via the EGFR pathway, providing insight into tumor cell proliferation and viability. miR-7 inhibits the EGFR pathway, decreasing cell invasiveness and viability. [40,46,47]	Glioblastoma
miR-34a	Functions such as survival, proliferation, apoptosis, invasion, migration, and stemness Regulates apoptosis, invasion and stem cell proliferation, and cell cycle arrest via Akt and Wnt pathways. Targets SIRT1 by regulating p53 expression. miRNA mimic triggers cell death in p53 mutant and chemo-resistant glioblastoma cell lines. [40,48,49,50].	Glioblastoma
miR-128	Functions such as proliferation, apoptosis, angiogenesis radioresistance, and stemness. Higher levels inhibit proliferation and stemness in vitro by regulating the Bmi-1 gene and E2F3a. Survival rates of the tumor increased further when along with the miR-128, other miRNAs were co-administered in GBM murine models. [40,51].	Glioblastoma

Abbreviations: Bmi-1: B lymphoma Mo-MLV insertion region 1 homolog, EGFR: Epidermal growth factor receptor, Bmi-1: B lymphoma Mo-MLV insertion region 1 homolog, SIRT1: Sirtuin 1, p53: Tumor protein p53, bcl-2: B-cell lymphoma 2, TGF- β: Transforming growth factor beta, HOX: Homeobox, EGFR: Epidermal growth factor receptor.

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
