# Peer review of "Exploring the Role of microRNAs in Glioma Progression, Prognosis, and Therapeutic Strategies"

_cancers, 2023, doi:10.3390/cancers15174213_

Round 1
Reviewer 1 Report
Review article on role of miRNAs in progression of gliomas is comprehensive overview of the current research on miRNAs. Article provides detailed role of specific miRNAs emphasizing their prognostic significance.
Few corrections in paper can be incorporated:
Line23: upregulation instead of upregulating
Line27-28: simplify the sentence
Figure1: Try to include miRNAs in WNT signaling pathway pictorial
Author Response
Point 1: Line23: upregulation instead of upregulating.
Response 1: Thank you for pointing this out. We have now rectified line 23 as requested.
Point 2: Line27-28: simplify the sentence.
Response 2: We appreciate your valuable input. We have carefully rectified lines 27-28 and simplified it to improve clarity and readability.
Point 3: Figure1: Try to include miRNAs in WNT signaling pathway pictorial.
Response 3: We greatly appreciate your suggestion to include miRNAs in the WNT signaling pathway pictorial in Figure 1. We agree that illustrating the role of miRNAs in this pathway will enhance the comprehensiveness of the figure, and we have included it accordingly.
Reviewer 2 Report
The text by Tluli, O., et. al., is interesting and well documented review on the relevant subject of Glial brain tumors. The concept of potential design of miRNAs for therapy is well explained in this text, I would only suggest the explanation on the title of “microRNAs” rather than “mi”, which could be a fair description for a general audience.
Author Response
Point 1: I would only suggest the explanation on the title of “microRNAs” rather than “mi”, which could be a fair description for a general audience.
Response 1: Thank you for your thoughtful feedback. We appreciate your suggestion, and we have included “microRNA” instead of “miRNA” as it can be more understandable for a broader audience.
Reviewer 3 Report
The submitted articles presents a concise review focusing on the role and potential usefulness of micro-RNAs (miRs) in gliomas. The manuscript includes three tables that summarize the current state of knowledge about miRs in gliomas. Most of the references are up to date and the manuscript is well written. However, there are some issues that shall be improved before the article could be considered for acceptance.
Reference 14 - check the authors
The title of article suggests that it focuses on the role of miRs in the progression and prognosis. However, the manuscript provides quite comprehensive reviow on potential utility of miRs in glioma treatment.
There is no information about the method for articles search and selection. The manuscript omits some recent works on miRs in gliomas and it could be useful for a reader to know the inclusion/exclusion criteria.
While the article does not focus on the role of long non coding RNAs in gliomas, their role and interaction with miRNAs shall be described at least in the introduction describing miRNAs and the ways they act on mRNA expression (chapter 2 and 3).
While the article focuses on miRNAs, theie contribution in the regulation of WNT pathway shall be presented in the Figure 1. Otherwise, the Figure shall be removed from the manuscript.
The title of Figure 1 is “The role of miRNAs in gliomas” while in the 3rd column it says “Expression in GBM as compared to normal”. Please correct “GBM” – some of the referenced articles were carried out on other gliomas.
The conclusion chapter (5.) shall be extended and, in addition to the conclusions, include short discussion summarizing the previous chapters and future perspectives.
The authors shall discuss more comprehensively perspectives for miRs in gliomas as diagnostic, prognostic or predictive markers, in addition to their usefulness and challenges to use miRs as treatment targets/treatment agents.
Lines 257-258 – lacks spacing between abbreviations and the body text.
Line 240: Please, check the numbering of sections/subsections – e.g. 3.3.1. miRs in therapy of gliomas (line 240) can’t be subsection for the 3.3. chapter (RB pathway).
Table 2 and Table 3 titles and some titles of sections/subsections – I recommend authors add “in gliomas” (when the referenced articles and duscission specifically refers to gliomas/glioblastomas etc.).
Section 3.3.1., lines 267-318 is chaotic, some paragraphs are unclear and it is generally difficult to follow for a reader. consider major revision of this section.
English Language is fine, minor editing and spelling correction recommended only.
Author Response
Point 1: Reference 14 – Check the authors.
Response 1: Thank you for bringing this to our attention. We apologize for the oversight in Reference 14. We have thoroughly reviewed the authors of the citation and have made the necessary changes.
Point 2: The title of article suggests that it focuses on the role of miRs in the progression and prognosis. However, the manuscript provides quite comprehensive reviow on potential utility of miRs in glioma treatment.
Response 2: Thank you for your thoughtful feedback. We understand your point that the title may suggest a specific emphasis on the role of miRNAs in glioma progression and prognosis, while the manuscript also covers the potential utility of miRNAs in glioma treatment. Accordingly, we have revised the title.
Point 3: There is no information about the method for articles search and selection. The manuscript omits some recent works on miRs in gliomas and it could be useful for a reader to know the inclusion/exclusion criteria.
Response 3: Thank you for your valuable suggestion. We agree that providing information about the method for article search and selection is essential for readers to understand the rigor and comprehensiveness of our review article. To address this, we have revised the manuscript to include a dedicated section outlining the methodology for the article search and selection process. In this section, we have provided a clear description of our research focus, the search strategy we used, as well as the specific inclusion and exclusion criteria applied.
Point 4: While the article does not focus on the role of long non coding RNAs in gliomas, their role and interaction with miRNAs shall be described at least in the introduction describing miRNAs and the ways they act on mRNA expression (chapter 2 and 3).
Response 4: Although it is not the focus of this review article, we do agree that providing some context about lncRNAs and their relevance to miRNA-mediated mRNA regulation would enhance the reader’s understanding of the broader regulatory networks involved in gliomas. Therefore, we have revised the introduction to incorporate a brief overview of the interplay of lncRNAs with miRNAs, from lines 105 – 110.
Point 5: While the article focuses on miRNAs, theie contribution in the regulation of WNT pathway shall be presented in the Figure 1. Otherwise, the Figure shall be removed from the manuscript.
Response 5: Thank you for your valuable feedback. We completely agree with the importance of highlighting the role of miRNAs in the regulation of the WNT pathway in Figure 1, and so we have included it accordingly.
Point 6: The title of Figure 1 is “The role of miRNAs in gliomas” while in the 3rd column it says “Expression in GBM as compared to normal”. Please correct “GBM” – some of the referenced articles were carried out on other gliomas.
Review 6: We highly appreciate your input in pointing this out. We apologize for the oversight on the title of the 3rd column of Table 1, as some of the references indeed pertain to other gliomas. We have revised and corrected this mistake.
Point 7: The conclusion chapter (5.) shall be extended and, in addition to the conclusions, include short discussion summarizing the previous chapters and future perspectives.
Response 7: Thank you for your valuable suggestion. We have carefully considered your feedback; however, we believe that our conclusion adequately summarized the key points discussed in the manuscripts. This includes the role of miRNAs in glioma prognosis and progression, highlighting specific upregulated and downregulated miRNAs. Furthermore, we have addressed how the pre-clinical studies done targeting miRNAs in gliomas hold potential for the future. With also keeping in mind that this is a review article, we have decided to not include a separate discussion, as it would reiterate the points already covered in the conclusion. We hope that this explanation is to your satisfaction.
Point 8: The authors shall discuss more comprehensively perspectives for miRs in gliomas as diagnostic, prognostic or predictive markers, in addition to their usefulness and challenges to use miRs as treatment targets/treatment agents.
Response 8: We sincerely appreciate your thorough evaluation of our manuscript. After careful consideration and deliberation, we acknowledge the significance of discussing the perspectives of miRNAs in gliomas extensively, and we firmly believe that our review article has effectively addressed this aspect. Your understanding is invaluable to us, and we remain open to any further feedback you may wish to share.
Point 9: Lines 257-258 – lacks spacing between abbreviations and the body text.
Response 9: We appreciate you bringing this issue to our attention. To enhance the reader’s experience, we have decided to relocate Table 2 and Table 3, along with their abbreviations, to the end of Section 4. By doing so, we aim to create a smoother transition from the abbreviations to the main text, making it more organized and thereby facilitating the reader’s navigation.
Point 10: Line 240: Please, check the numbering of sections/subsections – e.g. 3.3.1. miRs in therapy of gliomas (line 240) can’t be subsection for the 3.3. chapter (RB pathway).
Response 10: Thank you for pointing this out. we have made the necessary adjustment and renumbered this section as number 4, as it should not have been considered a subsection of 3.3.
Point 11: Table 2 and Table 3 titles and some titles of sections/subsections – I recommend authors add “in gliomas” (when the referenced articles and duscission specifically refers to gliomas/glioblastomas etc.).
Response 11: We appreciate your valuable suggestion to add “in gliomas” to the titles of Table 2 and Table 3, as well as to the titles of other sections in the manuscript. We have taken this into account and made the necessary changes.
Point 12: Section 3.3.1., lines 267-318 is chaotic, some paragraphs are unclear and it is generally difficult to follow for a reader. consider major revision of this section.
Response 12: Thank you for your feedback. Upon careful evaluation, we acknowledge that there were some issues with the organization and clarity of lines 267-318. In light of your feedback, we have taken action to improve the manuscript. As per your suggestion in Point 9, we have relocated Tables 2 and 3, along with their abbreviation list, to enhance the manuscript’s navigability and readability. Additionally, we simplified the title of that section and made necessary changes to certain parts of the text to ensure a smoother and more comprehensible reading experience for our audience. Once again, we appreciate your time and valuable insights.
Round 2
Reviewer 3 Report
The authors correctly addressed remarks of the reviewers. However, there are still some issues that shall be improved before the acceptance for publication:
1. the position of paragraph describing “Method for article search and selection:”. Please check if it follows the journal instruction, otherwise I suggest to transfer it after the Introduction.
2. Figure 1 - please present interactions beteween the specific miRNAs and their targets belonging to the WNT pathway in the Figure. The Figure presents mainly the WNT pathway but not specific miRNAs and it shall be corrected or removed.
Author Response
Point 1: the position of paragraph describing “Method for article search and selection:”. Please check if it follows the journal instruction, otherwise I suggest to transfer it after the Introduction.
Response 1: Thank you for your suggestion. We have reevaluated the placement of the paragraph “method for article search and selection” and subsequently relocated it to follow the introduction.
Point 2: Figure 1 - please present interactions beteween the specific miRNAs and their targets belonging to the WNT pathway in the Figure. The Figure presents mainly the WNT pathway but not specific miRNAs and it shall be corrected or removed.
Response 2: Thank you for your valuable feedback regarding Figure 1. While the WNT pathway’s representation in the figure is relevant to the context of the article, we understand that the absence of specific miRNA interactions with their targets may limit the figure’s overall utility and clarity. We recognize the significance of this information, and we believe that presenting this complex network of interactions in a single figure may not effectively convey the detailed mechanisms involved. Therefore, after careful consideration, we have decided to remove the figure from the manuscript, while focusing on presenting this valuable information in the main text as it is currently.